# Ecological and social constraints combine to promote evolution of non-breeding strategies in clownfish

Rebecca Branconi [1✉], Tina A. Barbasch [1], Robin K. Francis[1], Maya Srinivasan[2], Geoffrey P. Jones [2] & Peter M. Buston [1]

Individuals that forgo their own reproduction in animal societies represent an evolutionary paradox because it is not immediately apparent how natural selection can preserve the genes that underlie non-breeding strategies. Cooperative breeding theory provides a solution to the paradox: non-breeders benefit by helping relatives and/or inheriting breeding positions; non-breeders do not disperse to breed elsewhere because of ecological constraints. However, the question of why non-breeders do not contest to breed within their group has rarely been addressed. Here, we use a wild population of clownfish (*Amphiprion percula*), where non-breeders wait peacefully for years to inherit breeding positions, to show non-breeders will disperse when ecological constraints (risk of mortality during dispersal) are experimentally weakened. In addition, we show non-breeders will contest when social constraints (risk of eviction during contest) are experimentally relaxed. Our results show it is the combination of ecological and social constraints that promote the evolution of non-breeding strategies. The findings highlight parallels between, and potential for fruitful exchange between, cooperative breeding theory and economic bargaining theory: individuals will forgo their own reproduction and wait peacefully to inherit breeding positions (engage in cooperative options) when there are harsh ecological constraints (poor outside options) and harsh social constraints (poor inside options).

[1] Department of Biology, Boston University, 5 Cummington Mall 101, Boston, MA 02215, USA. [2] ARC Centre of Excellence for Coral Reef Studies, and College of Science & Engineering, James Cook University, Townsville 4811 QLD, Australia. ✉email: branconi@bu.edu

The evolution of non-breeding and cooperative behaviors, and the formation of social groups, can be readily understood using Hamilton's inequality[1,2] (Fig. 1). Individuals will be more likely to forego their own reproduction and engage in cooperative behaviors, if there is high relatedness between group members[2,3] such that they can pass on their genes by helping their relatives in the present[4–6] and/or if there is a high probability of inheriting a breeding position[7,8] such that they will pass on their genes in the future[9–11] (i.e., the left hand side of the *Hamilton's inequality* is high). Also, individuals will be more likely to forego their own reproduction and engage in cooperative behaviors, if there are strong ecological constraints[12,13] such that there are no opportunities for breeding outside of the group[14–16] and/or if there are strong social constraints such that there are no immediate opportunities for breeding inside the group[17–19] (i.e., the right hand side of the *Hamilton's inequality* is low). While there is extensive observational and experimental evidence demonstrating that high relatedness, future benefits and ecological constraints help explaining non-breeding behaviors in animal societies, relatively limited work has been done to investigate the roles of social constraints[20]. Furthermore, there is a real need to broaden the diversity of social taxa and types of cooperative behaviors considered, so that we may better understand the drivers of social group formation across taxa and along the continuum from simple to complex eusocial systems[21,22].

The clown anemonefish (*Amphiprion percula*) lives in social groups composed of a breeding pair and zero to four non-breeding individuals on the coral reefs of Papua New Guinea[23]. Non-breeders cooperate by remaining small and not inflicting costs on their dominants[24,25], but why they engage in such peaceful cooperation remains untested[25,26]. Group members are not related[27] and non-breeders do not provide alloparental care[28] but they do inherit the territory within which they reside following the death of the breeders[29]. Each group is confined to a sea anemone (*Heteractis magnifica*) that affords protection from predators[30–33]. However, every anemone of the reef is occupied[32,34,35], because there is high recruitment rate (due to a constant rain of larval settlers that disperse from their natal anemones from distances up to 120 km[35,36]), and low mortality rate[35,37,38]. In addition, it is risky to move between anemones, because clownfish are poor swimmers and can be preyed upon[30,31,33,39]. Taken together, habitat saturation and risks of

movement likely reduce the payoff associated with leaving to breed elsewhere, suggesting that ecological constraints play a role in social group formation. Within each group there is a size-based dominance hierarchy[24] where the female is the largest (rank 1), the male is second largest (rank 2) and the non-breeders get progressively smaller (ranks 3–6); if the female of the group dies, then the male changes sex and becomes the new female (clownfish are protandrous hermaphrodites[23,40]), and the largest non-breeder becomes the new male[29]. Within the size hierarchy subordinates tend to be 80% of the size of their immediate dominants[41]. This factor likely reduces the payoff associated with contesting for breeding positions, suggesting that social constraints also play a key role in social group formation.

The aim of this study is to investigate why clownfish nonbreeders engage in the cooperative option, waiting peacefully in social groups to inherit breeding positions, rather than engaging in one of two, alternative, non-cooperative options: (i) the outside option—leaving to breed elsewhere; and ii) the inside option—contesting to breed at home (Fig. 1).

## Results

**Ecological constraints experiment #1**. To test the hypothesis that non-breeding individuals do not disperse to breed elsewhere because of strong ecological constraints in the form of risk of mortality during dispersal, we experimentally tested the critical prediction that non-breeding individuals will disperse when the risk of moving between anemones is reduced. Risk was to be manipulated by presenting alternative anemones in succession at a distance of 0.5 m and 5.0 m from 32 focal groups (Fig. 2a). To explore the effect of variation in the alternative option, two classes of anemones were used: empty anemones ($n = 16$) or anemones with a breeding male significantly larger than the focal nonbreeder ($n = 16$). Focal anemones were assigned to one of the two options at random. The two classes of anemone represent different potential outcomes for the focal non-breeder: if it were to disperse to the empty anemone, it would become the breeding female, and would have to wait for a new recruit to breed; if it were to disperse to the anemone with a breeding male, it would become the breeding male, and would have to wait for the resident breeding male to change sex to breed. Focal groups all had at least one non-breeder. We left alternative anemones alongside the home anemones for 2 days, to allow the focal non-breeder

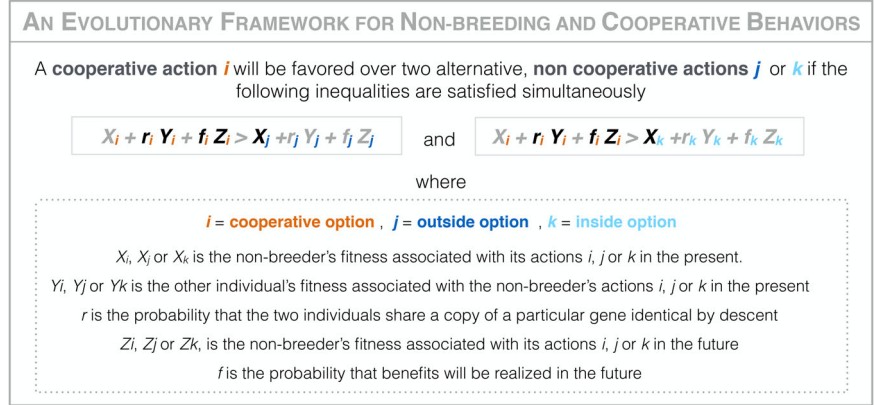

**Fig. 1 An evolutionary framework for non-breeding and cooperative behaviors based on Hamilton's inequality.** The evolution of non-breeding and cooperative behaviors depends on the expected fitness associated with engaging in non-breeding and cooperative actions (the cooperative option) relative to alternative non-cooperative actions outside the group (the outside option) or inside the group (the inside option). The cooperative option can be favored by selection because of its beneficial effects on kin (e.g., via helping relatives) and in the future (e.g., via territory inheritance) and because of the low expected fitness associated with the outside option (e.g., due to ecological constraints) and the inside option (e.g., due to social constraints). In general, when we are trying to explain the evolution of non-breeding strategies, $X_i$ on the left hand side and $r_j Y_j, f_j Z_j, r_k Y_k, f_k Z_k$ on the right hand side (terms in light gray) are considered to be zero or trivially small compared to other terms (terms in black).

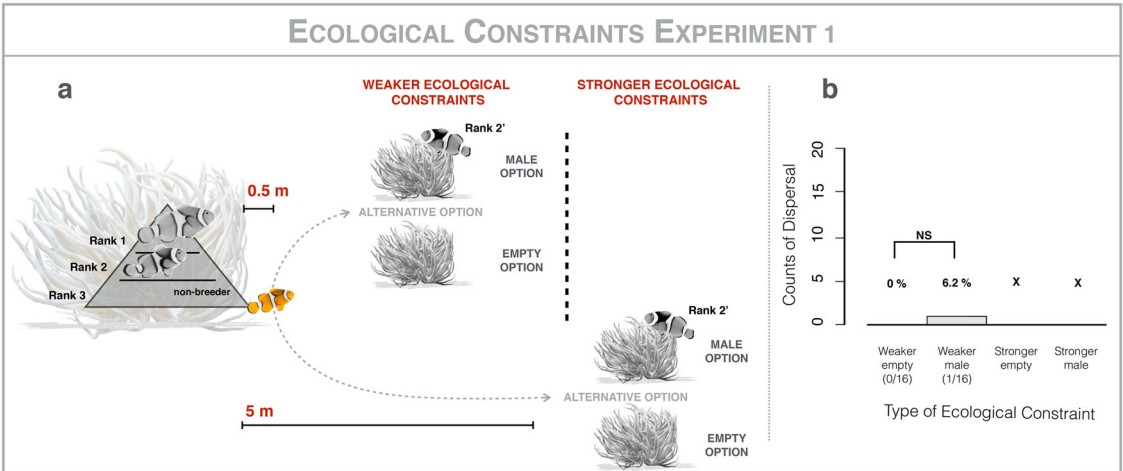

**Fig. 2 Ecological constraints experiment #1. a** Methods: presentation of two classes of alternative anemones (empty anemones, $n = 16$; anemones with a breeding male, $n = 16$) at a distance of 0.5 m (weaker ecological constraints) and 5 m (stronger ecological constraints) from 32 focal groups. **b** Results: comparison of the counts of dispersal of rank 3 non-breeding individuals by type of ecological constraint: weaker-empty; weaker-male; stronger-empty; stronger-male; because so few non-breeders moved to the alternative anemone at 0.5 m, we did not present the alternative anemones at 5 m, and this is denoted by "X"; Fisher's Exact Tests: "NS" non-significant ($p > 0.05$); "*" significant ($p < 0.05$); horizontal brackets indicate pairwise comparisons. All work was conducted using *A. percula* groups inhabiting *H. magnifica* anemones, on inshore reefs near Mahonia Na Dari Research and Conservation Centre, in Kimbe Bay, Papua New Guinea, from June to September 2018, using SCUBA at depths of up to 20 m. At the end of the experiment, all individuals were returned to their home anemones.

sufficient time to make a choice. The morning of the third day, we recorded whether focal non-breeders (rank 3) had moved to the alternative anemone. Focal non-breeders dispersed to the alternative option placed at 0.5 m in only one out of 32 cases (Fig. 2b). Because so few non-breeders moved even 0.5 m and because (i) studies on fish with similar social systems indicated that likelihood of movement declined rapidly as a function of distance[42] and (ii) previous studies on clownfish have demonstrated that they don't move greater distances in response to naturally occurring vacancies[29,35], we did not present the 5.0 m option. Our result supports the hypothesis that non-breeders do not disperse to breed elsewhere because of the risks associated with moving even short distances. However, this result could also provide support for two alternative hypotheses: non-breeders do not disperse because their home anemone confers higher expected reproductive success than the presented alternative, possibly because there are some benefits in stable cooperative relations with other fish within the group or stable mutualistic relationship with the anemone[43]; non-breeders do not disperse because there is limited plasticity of movement in clownfish (i.e., moving from their home anemone is not in their behavioral repertoire), just as in many[44], but not all[45,46], social insects.

**Ecological constraints experiment #2.** To discriminate among these three alternative hypotheses, we adjusted the experimental design, and experimentally tested the critical prediction that non-breeders will not return to their home anemone when the risk of moving between anemones is increased. We presented alternative anemones at a distance of 0.5 m and 5.0 m from 32 focal groups. Each focal group was tested for both distances, in series. As above, we started with the 0.5 m experiment because if there were no movement there, then we would not predict any movement to 5.0 m[29,35,42]. Once more, two classes of anemones were used at each distance: empty anemones ($n = 16$) or anemones with a breeding male ($n = 16$). In this case, however, we relocated the focal non-breeder from the home anemone to the alternative (Fig. 3a). We left this set-up for 2 days, to allow the focal non-breeder sufficient time to make a choice. The morning of the third day, we recorded whether the focal non-breeder had returned to

its home anemone. When the alternative option was placed at 0.5 m, the focal non-breeder returned to its home anemone in 22 out of 32 cases (13/16 from the empty anemone; 9/16 from the anemone with a breeding male; Fig. 3b). This result rejects the hypothesis that there is limited plasticity of movement in clownfish (i.e., movement between anemones is in their behavioral repertoire even though its rarely seen under natural conditions). Notably, there were significantly more movements in this experiment than in the first experiment (Fisher's exact test, $p < 0.001$). This result supports the hypothesis that non-breeders did not disperse in the first experiment because their home anemone confers higher expected reproductive success than the alternative, though it also suggests that there may be some risk to movement even in the 0.5 m treatment because not all focal non-breeders returned home in the second experiment. Finally, when the alternative anemone was presented at 5.0 m, non-breeders returned to their home anemone in zero out of 32 cases (Fig. 3b)—in all cases, fish remained inside the alternative anemones. This is significantly less than in the 0.5 m case (Fisher's exact test, $p < 0.001$). Given that anemones tend to be tens of meters apart under natural conditions, this result supports the hypothesis that non-breeders do not disperse to breed elsewhere because of harsh ecological constraints in the form of risks of movement.

**Social constraints experiment.** To test the hypothesis that non-breeding individuals do not contest for breeding positions because of strong social constraints in the form of evictions of non-cooperative individuals, we experimentally tested the critical prediction that non-breeding individuals will contest for breeding positions when the probability of winning a contest is increased. To test this prediction, we used 16 focal groups, all of which consisted of at least three individuals and had bred at least once in the preceding two months (Fig. 4a). All individuals in each focal group were caught and measured to the nearest 0.1 mm using calipers, and the largest non-breeder (rank 3) was removed. Then, we introduced two types of rank 3 individuals to the focal group: a non-breeder less than 80% of the size of the breeding male (rank 3') or a non-breeder more than 80% of the size of the

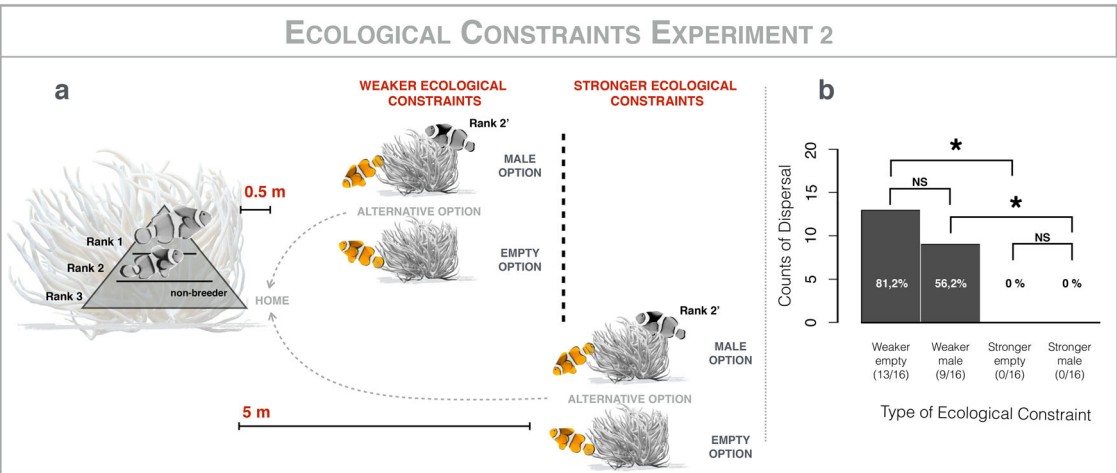

**Fig. 3 Ecological constraints experiment #2. a** Methods: relocation of rank 3 non-breeding individuals inside one of the two alternative classes of anemones (empty anemones, $n = 16$; anemones with a breeding male, $n = 16$) at a distance of 0.5 m (weaker ecological constraints) and 5 m (stronger ecological constraints) from 32 focal groups. **b** Results: comparison of the counts of dispersal of rank 3 non-breeding individuals by type and strength of ecological constraint: weaker-empty; weaker-male; stronger-empty; stronger-male; Fisher's Exact Tests: "NS" non-significant ($p > 0.05$); "*" significant ($p < 0.05$); horizontal brackets indicate pairwise comparisons. All work was conducted using *A. percula* groups inhabiting *H. magnifica* anemones, on inshore reefs near Mahonia Na Dari Research and Conservation Centre, in Kimbe Bay, Papua New Guinea, from June to September 2018, using SCUBA at depths of up to 20 m. At the end of the experiment, all individuals were returned to their home anemones.

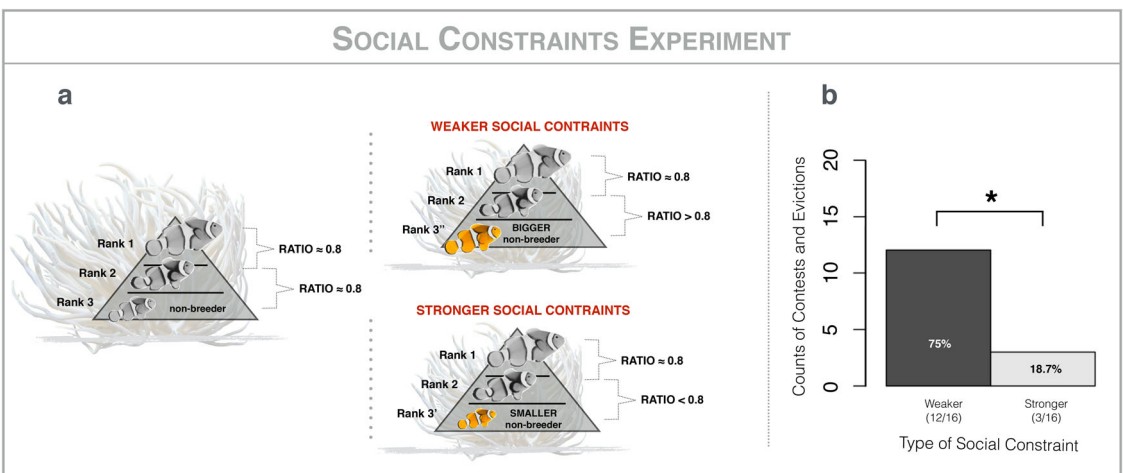

**Fig. 4 Social constraints experiment. a** Methods: removal of rank 3 non-breeding individuals from 32 focal group and introduction of rank 3 of a different size, either a few millimeters smaller (R3'; stronger social constraints; $n = 16$) or a few millimeters bigger (R3"; weaker social constraints; $n = 16$) than the original rank 3. **b** Results: comparison of the counts of contests by strength of social constraints; Fisher's Exact Tests: NS" non-significant ($p > 0.05$); "*" significant ($p < 0.05$); horizontal brackets indicate pairwise comparisons. All work was conducted using *A. percula* groups inhabiting *H. magnifica* anemones, on inshore reefs near Mahonia Na Dari Research and Conservation Centre, in Kimbe Bay, Papua New Guinea, from June to September 2018, using SCUBA at depths of up to 20 meters. At the end of the experiment, all individuals were returned to their home anemones.

breeding male (rank 3"). The introduced rank 3 individuals, hereafter called introducees, always came from an anemone other than the focal anemone. Each focal group received rank 3' and rank 3", one at a time, on different days, in random order. We left introductions overnight and, the following day, we noted whether there had been a contest, as indicated by an eviction—evictions are a common outcome of contests in this system[47]. If introducees were not found in the focal anemone the following day, they were considered evicted because i) they do not leave voluntarily and ii) mortality without eviction is rare[35]. Such individuals were classified as disappearances if they could not be found in any of the other anemones on the reef after careful inspection[35]. If introducees were still present, they were considered evicted if they spent more than 3 min out of 5 outside the

anemone (i.e., with their full body length outside of the range of anemone tentacles) and considered tolerated if they were still present and spent less than 3 min outside of the anemone[48]. When the introduced non-breeder was less than 80% of the size of breeding male (rank 3'), it contested for a breeding position and was evicted in only 3/16 cases and it was tolerated in 13/16 cases. When the introduced non-breeder was greater than 80% of the size of the breeding male (rank 3"), it contested for a breeding position and was evicted in 12/16 cases and it was tolerated in 4/16. This difference was significant (Fisher's exact test; $p = 0.004$; Fig. 4b). Of the 15 rank 3 non-breeders evicted, 7 remained in the vicinity of the focal anemone but 8 had disappeared—this result demonstrates that the threat of predation post eviction from an anemone is real. Given that non-breeders are ~80% of the size of

their immediate dominant under natural conditions[41] this result supports the hypothesis that non-breeders do not contest for breeding positions because of harsh social constraints.

## Discussion

Our findings explain why clownfish non-breeders forgo their own reproduction, resolving this evolutionary paradox. While they do not gain indirect genetic benefits from helping kin[28], they do stand to gain future direct benefits by inheriting a breeding position on the death of a breeder[29]. Here, we show that they tolerate their non-breeding situation because harsh ecological constraints, in the form of habitat saturation and risks of movement, prevent them from successfully dispersing to breed elsewhere. Further, we show that they tolerate their non-breeding situation because harsh social constraints, in the form of well-defined size differences between individuals adjacent in rank, prevent them from successfully contesting to breed at home. Compellingly, we show that individuals will disperse and contest to better their current situation when ecological and social constraints are experimentally relaxed. In clownfish, ecological and social constraints combine to promote the evolution of non-breeding strategies, and it is necessary to understand both types of constraint to fully understand their societies.

A striking result of our first and second ecological constraints experiments is that non-breeders did not leave home when presented with an alternative anemone at 0.5 m and two-thirds of them returned home when relocated to an alternative anemone at 0.5 m. This strongly suggests some benefit of stable associations with known anemones of particular qualities, with familiar fish with whom conflicts have been resolved, and/or with a larger group of fish. Anemones vary in size and expansion behavior, which impacts the safe foraging area available to the fish, which in turn influences the size of the fish, their investment in egg laying and parental care, and their reproductive success[49,50]. A larger group of fish may enhance the size of their anemone, either by providing more nutrients enabling the anemone to grow more[51–53], or by defending against anemone predators allowing anemones to expand their tentacles and feed more[53]. Taken together, these factors could explain why non-breeders have such a strong affinity for their home anemones and the associated fish, and future studies will test these alternative hypotheses.

Another striking result of the second ecological constraints experiment and our social constraints experiment is that not all individuals responded in the same way to the manipulations: some individuals returned home while others did not; some individuals contested while others did not. One interesting hypothesis is that variation in response could be due to variation in personality traits (defined as inter-individual differences in behavior that are consistent over time and across contexts[54]) of the individuals being tested. Previous studies have demonstrated the existence of personality traits (e.g., boldness, aggressiveness, shyness, sociability, and parenting behaviors) in *A. percula*[55,56] and its sister species *A. ocellaris*[57]. It was not possible for us to test this idea, or alternative explanations for the variation in response, with our data. Future studies will investigate how individual personality traits influence individuals' decisions in clownfish societies, and how different combinations of individuals' personality traits influence the form and function of clownfish societies.

In sum, here, we have shown that ecological and social constraints combine to promote the evolution of non-breeding strategies and the formation of complex social groups. Interestingly, this explanation for social group formation can be framed in the language of economic bargaining theory[58–60]. Economic bargaining theory emphasizes that there are three options available to individuals: the cooperative option (the payoff from pursuing cooperative actions inside the group), the outside option (the payoff from pursuing non-cooperative actions outside of the group[58–60]) and the inside option (the payoff from pursuing non-cooperative actions inside the group[58–60]). Individuals will engage in cooperative actions when both the outside option and the inside option are poor relative to the cooperative option. Therefore, synthesizing the language of economic bargaining theory and cooperative breeding theory, reveals that individuals will forgo their own reproduction, engaging in cooperative actions such as remaining small and waiting peacefully to inherit territories, when there are poor options outside the group (strong ecological constraints) and poor options inside the group (strong social constraints). The synthesis of these two theories may lead to a fruitful exchange of ideas between fields and advances in our understanding of complex societies.

## Methods

**Study population.** We studied the clown anemonefish *Amphiprion percula* in Kimbe Bay, Papua New Guinea, from June to September 2018. All work was conducted using SCUBA at depths up to 20 meters. We located 186 magnificent sea anemones (*Heteractis magnifica*) on 12 inshore reefs near Mahonia Na Dari Research and Conservation Centre. Each anemone was occupied by a single group of *A. percula*. Groups consisted of a breeding pair and zero to three non-breeders. Individuals were identified based on natural variation in their color markings.

**Anemone acquisition.** We surveyed the anemone population and identified a small number of anemones that were movable e.g., attached to small rocks or only loosely attached to the hard substrate. These anemones were collected and used as alternative anemones for the ecological constraints experiments. For these experiments, alternative anemones were placed at a distance of 0.5 m or 5.0 m from the focal anemones with a clear line of sight between the two. Fish have been shown to locate anemones using chemical and visual cues at these distances[32,33,61,62]. [During the experiment, unused fish from the alternative anemones were kept in the laboratory at Mahonia Na Dari Research and Conservation Center; at the end of the experiment, all fish and anemones were returned to their original location].

**Fish measurement.** In June, we captured all fish using hand nets, placed them inside clear plastic bags, laid them against a slate, and measured their standard length to the nearest 0.1 mm using calipers. This entire procedure was conducted underwater and all individuals were returned to their anemone within a few minutes. Individuals were ranked (1–5) based on their size relative to other individuals within the anemone, with the largest being ranked 1. Rank 1 was designated as the female, rank 2 the male, and ranks 3–5 as non-breeders. We monitored the fish population to determine which groups were breeding. Breeding was readily detectable because the male spends much of his time caring for the eggs.

**Statistics and reproducibility.** For the first ecological constraints experiment, to test the hypothesis that the likelihood of rank 3 non-breeders dispersing will depend on the classes of alternative anemones (empty anemones or anemones with a breeding male), we used one Fisher's exact test for contingency tables. Specifically, at a distance of 0.5 m, we tested whether the number of rank 3 non-breeders that dispersed from their focal anemone to empty anemones ($n = 16$) differed from the number of rank 3 non-breeders that dispersed from their focal anemone to anemones with a breeding male ($n = 16$).

For the second ecological constraints experiment, to test the hypothesis that likelihood of rank 3 non-breeders returning home will depend on the classes of alternative anemones and their distances from the focal anemones, we used four Fisher's exact tests for contingency tables. First, at a distance of 0.5 m, we tested whether the number of rank 3 non-breeders that returned home from empty anemones ($n = 16$) differed from the number that returned home from anemones with a breeding male ($n = 16$). Second, we conducted an equivalent test at 5.0 m ($n = 16$ for each treatment). Third, for alternative anemones that were empty, we tested whether the number of rank 3 non-breeders that returned home from 0.5 m ($n = 16$) differed from the number that returned home from 5.0 m ($n = 16$). Fourth, we conducted an equivalent test for alternative anemones with a breeding male ($n = 16$ for each treatment).

For the social constraints experiment, to test the hypothesis that bigger introducees (rank 3", weaker social constraints), but not smaller introducees (rank 3', stronger social constraints), will contest for breeding positions and will be evicted by the breeding pair, we used one Fisher's exact test for contingency tables. Specifically, we tested whether the number of rank 3" that were evicted ($n = 16$) differed from the number of rank 3' that were evicted ($n = 16$).

All analyses were done in R v. 3.4.2 'Short Summer'.

**Reporting summary**. Further information on research design is available in the Nature Research Reporting Summary linked to this article.

## Data availability
The datasets analyzed during the current study are available in the Dryad Digital Repository (https://doi.org/10.5061/dryad.sf7m0cg49) and from the corresponding author on reasonable request.

## Code availability
The full codes analyzed during the current study are available in the Dryad Digital Repository (https://doi.org/10.5061/dryad.sf7m0cg49) and from the corresponding author on reasonable request.

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

## Acknowledgements

This manuscript forms part of R.B. doctoral dissertation requirements (Boston University). We are particularly grateful to James Traniello, Stephen Emlen, and Peter Nonacs for constructive feedback on earlier versions of the manuscript. We would also like to thank M. Schniedewind for assistance in the field, the staff of Mahonia Na Dari Research and Conservation Centre and Walindi Plantation Resort for logistical support in the field, and the communities of Tamare and Kilu, the traditional owners of the reefs. All work was performed with the approval of the Institutional Animal Care and Use Committee, Boston University (Protocol number: 17/001) and the Government of Papua New Guinea. The research was supported by one Sigma XI Grant-in-Aid of Research and one Kunz award awarded by Boston University to R. Branconi and by one NSF Doctoral Dissertation Improvement grant (grant number: IOS- 1701657), one Warren McLeod fellowship and one BU Women's Guild award awarded by Boston University to T. Barbasch.

## Author contributions

R.B., T.A.B., and P.M.B. participated in research design. R.B., T.A.B., and R.K.F performed data collection, using the study population established by M.S. and G.P.J. R.B. and T.A.B. conducted data analysis. R.B., T.A.B., R.K.F., M.S., G.P.J., and P.M.B. wrote the manuscript. All authors approve the final version of the manuscript, and agree to be held accountable for the content therein and declare no competing interests.

## Competing interests

The authors declare no competing interests.
