## [Peer Review File · Communications Biology]

Reviewers' comments:

Reviewer #1 (Remarks to the Author):

This is nice study on what maintains size hierarchies in cooperative groups of clownfish, and shows that it is rather difficult to entice fish to move between anemones. The authors conclude that this results from strong social constraints within anemones, predation risk, and habitat saturation. I think, however, that their experiments more so implies these factors than definitively demonstrating them.

Predation risk:

- The authors cite only one study that purports that it is dangerous for clownfish to leave their anemone. Was there an actual predation rate measured in this work? If so, it would be useful to give these data. It seems that authors may be assuming that because you don't see a lot of clownfish swimming freely, it is dangerous to do so. It would be certainly implied that it is dangerous, but not actually conclusively demonstrated to be true.
- Within the experiment, there seems to be no predation risk. Fish do not disappear from the targeted groups (which might be expected if some fish that try to switch, got eaten while moving). None of the fish that were experimentally moved, disappeared. Thus, every fish that tried to get back to its home anemone, succeeded. Even transplants that were ejected, could apparently hang around outside of anemones. This, of course, doesn't prove that there is little or no predation risk, but lack of ANY reported predation during the length of this study does suggest the risk might not be that extreme.

Habitat saturation:

- The authors initially state that more or less every anemone is 'full' of fish. Thus, what you have under normal conditions is a very flat market. Even without any predation risk, it just may never pay off to search the habitat and risk losing one's place in rank at home. I.e., you can't ever really trade up from your 'starter home'.
- Does a fish even notice that there is an empty anemone 0.5 or 5m away? The 0.5m distance would seem close enough, but fish might not even realize that a better anemone is 5m away. Therefore, they do not realize that a choice exists.
- As a very interesting comparison, I suggest the authors look at a pair of papers by Lena Grinsted and Jeremy Field in 2017 on paper wasps, where they asked a very similar question of if subordinates will move to increase their fitness. They found subordinates do move (wasps seem more inclined to check out neighbors than clownfish), but in most cases fitness is not increased. Again, this is due to flat market, where the choices are probably very close in payoff.

Social constraint:

My only concern here is in the verbiage used. Transplants that disappeared were claimed to be contesting for rank, and therefore ejected. The authors, however, have no data on what happened – just that fish were gone. Rather than "contest" (Lines 148-152), maybe a better phrasing is that they were "not tolerated" or "not accepted", as there is no indication as to how the transplants behaved. Or maybe, they just left to try to get back to their original anemone?

As a minor point, clownfish were compared to social insects as species where subordinates tend to stay at home (Line 98). However, workers in many species do very often drift between nests to check them out or even join them (see Nonacs, 2017, *Frontiers Ecol. & Evol.*). They seem much less attached to their home than clownfish!

Respectfully submitted,

Peter Nonacs

Reviewer #2 (Remarks to the Author):

This is a well-written article on an interesting and important subject in evolutionary ecology. The authors take advantage of a great study system and use a set of experimental manipulations to make inferences.

I struggled to understand some of the key elements of the experimental designs. No doubt this is a by-product of the short format, but I think some modest elaboration is warranted. Without those details, the strict interpretation of some analyses is uncertain for the reader. The decision to not conduct a set of treatments in the first experiment was unfortunate, and raises some uncertainties around the overall interpretations.

Specific comments (in descending order of importance):

1. L 87-89: I think this was an unfortunate decision, and it is frustrating for the reader. I spent a fair bit of time trying (and failing—because key details were missing) to understand the nature of these treatments, only to learn here that they were never actually implemented. While I don't dispute that movement across the longer distance is unlikely given most fish didn't move much across the shorter distance (and they didn't move 5m in Experiment 2), I think this decision represents a lapse in an otherwise elegant design. Had those trials been conducted, and no fish moved 5m in Experiment 1, the authors' inferences would have been greatly strengthened. However, had ANY of those individuals moved, this would have necessitated a substantially different interpretation. The problem is, we cannot know what would have happened, and so we must base inferences on an untested assumption (frustratingly, these treatments were originally planned, but not executed). The other issue I have with this is that it de-values the importance of robust experimental design (and serves as a poor example for others who may seek exemplars from high-impact journals). I don't think this needs to be a deal-breaker for publication, but I think some acknowledgment of the issues is warranted.

2. I was unable to follow some of the experimental designs, even allowing for information split across the main text, figure captions, and supplementary methods. Were focal anemones assigned to treatments at random? Were the same anemones/fish evaluated for both distances (in the experiment where this was done)? If so, were these done in series; was there an interval between repeated trials; was the order of presentation (of distances) randomised across focal individuals? Is there any indication that individuals' responses were correlated across trials (bold/shy phenotypes)? Is there any possibility that any of these affected the outcomes of trials? In experiment 3, were fish cleared from anemones and then reassigned to other anemones, or was a completely new sample of 'rank 3' fish collected and used? (I ask because I wonder if precautions were taken to ensure that focal fish were not inadvertently back-transplanted to their anemone/social group of origin—which would confound the study and change my interpretations).

3. I think the presentation of statistical results (and their interpretation) may imply (at least to some readers) that some experimental designs were comprised an orthogonal set of treatments and independent observations. If fish were re-used, this isn't the case. Figure legends (and/or Supplementary Methods) could be modified to clarify the design and the nature of the Fisher's exact tests. In Fig 2b. the legend should more clearly indicate that the horizontal brackets indicate pairwise comparisons (as opposed to some sort of nested/hierarchical analysis—at first glance, this is how the figure appeared to me).

4. L46-68: This paragraph has a structure that makes some of its components seem redundant. I suspect some minor revision could address this, to streamline the content (if necessary) and allow for more description of the breeding system (as per my next comment).

5. L 79-80: Some explicit predictions are offered, and to understand these, the reader needs to

know about sex change in this species. This isn't explicitly covered until later. I think the breeding system could be described in the preceding paragraph (L 46-68).

6. L 81-82: Is this certain? (e.g., because a certain size-structure was imposed by the experimental design?) The rationale for this statement isn't clear.

7. Line 141 and elsewhere: Consider whether the use of the term "Introducees" might cause some confusion for non-native English readers.

8. In several instances, "Individuals" (with a capital "I") is used; in others it is lower case. I think there may be some meaning ascribed to this, but I found it unnecessary and a distraction.

9. Line 48-49: "...avoiding inflicting..." is slightly awkward

I supply these comments in the hope that they may enable the authors to strengthen their contribution. I want to be clear that I really liked this manuscript; I think it is a worthy contribution, and I look forward to seeing it published.

Reviewer #3 (Remarks to the Author):

This manuscript presents the results from a set of clever and careful experiments to determine why clownfish individuals queue to wait for breeding opportunities rather than challenge for a breeding position in their own or another group. I think the data contribute important insights, the analyses are sufficient, and the findings are placed in the broader context. I have one question about the interpretation of one conclusion though, and a few suggestions for potential edits that might help readers to better understand the findings.

The interpretation I am wondering about whether it is indeed 3rd ranking males that initiate a contest or whether it is the male that would be challenged. In your experiment where you place an individual larger than 80% of the body size in a group, you argue that 12/16 of these "contested for a breeding position" (line 151). You interpret this finding saying that "we show that individuals will disperse and contest to better their current situation when ecological and social constraints are experimentally relaxed" (Line 166f). You assume that the introduced 3rd ranked individuals contest when they get closer in size - wouldn't a more parsimonious explanation be that the male starts attacking the 3rd ranking individual once it get's too close in size? I think this is an important distinction: while lower ranking individuals seem to be able to influence their growth, I am not sure they can ever switch to actively contest for that position. Whenever they grow to much, the male will start attacking and evicting them. As your results show (line 149f), these 3rd ranking individuals never win. Accordingly it is not a true contest, it rather seems that males feel challenged and evict. This also makes me wonder how comparable this situation is to the economic bargaining theory that you describe: in it, actors appear to have a choice because individuals seem to be relatively similar. In the clownfish, there are inherent differences among individuals that limit the options individuals might be able to take. Do you have other data that might put these conclusions in context? For example, is contest based on a probabilistic assessment, where the closer in body size 3rd ranking individuals are the more likely they will be attacked?

Some additional information that might be helpful:

You mention that there is a "high recruitment rate" (Line 55): it might help to have more

information on how recruitment occurs (if you have it): is it larval and they crawl, do individuals float and randomly settle on the first anemone they find, or do they swim within a certain radius?

For the individuals displaced 5 meters to a new anemone, you say that none returned to their home anemone (line 121): what happened to them? Did they move at all? Did they die?

Are dominant males/females all of the same size, independent of which anemone they are in? If there is variation, would a 3rd ranking individual from a large anemone be able to outcompete a male/female from a smaller anemone? But this might not occur because, as you show, there is some form of home advantage (knowing the place, the other individuals) or because their own fitness will be so much larger on the better anemone that it makes up for waiting?

Reviewers' comments:

Reviewer #1 (Remarks to the Author):

This is nice study on what maintains size hierarchies in cooperative groups of clownfish, and shows that it is rather difficult to entice fish to move between anemones. The authors conclude that this results from strong social constraints within anemones, predation risk, and habitat saturation. I think, however, that their experiments more so implies these factors than definitively demonstrating them.

Predation risk

- The authors cite only one study that purports that it is dangerous for clownfish to leave their anemone. Was there an actual predation rate measured in this work? If so, it would be useful to give these data. It seems that authors may be assuming that because you don't see a lot of clownfish swimming freely, it is dangerous to do so. It would be certainly implied that it is dangerous, but not actually conclusively demonstrated to be true.

Thank you for encouraging us to clarify this important point. Previous works shown that taking clownfish from their anemones or removing anemones from beneath their fish quickly results in the clownfish becoming prey of larger fish (e.g. Verwey 1930 *Treubia*, Marsical 1970 *Mar. Biol.*; Fautin Symbiosis 1991, Elliot et al, *Mar. Biol.* 1995). In the revised version of the manuscript we added the citation for all these studies line 54). Because the results of these studies are so unambiguous we considered it unethical to repeat them to assess predation risk.

- Within the experiment, there seems to be no predation risk. Fish do not disappear from the targeted groups (which might be expected if some fish that try to switch, got eaten while moving). None of the fish that were experimentally moved disappeared. Thus, every fish that tried to get back to its home anemone, succeeded. Even transplants that were ejected, could apparently hang around outside of anemones. This, of course, doesn't prove that there is little or no predation risk, but lack of ANY reported predation during the length of this study does suggest the risk might not be that extreme.

In contrast to predation risk outside of anemones, predation risk inside of anemones is very low (Buston 2003 *Mar Biol*; Buston & Garcíá 2007 *J Fish Biol*). While we did not record any deaths of clownfish during the first and second experiments when the fish were residing safely in an anemone or making a move between anemones at a time of their own choosing, we recorded a high rate of disappearance of the transplants that were forcibly evicted in the third experiment. In this case, in 8 out of 15 cases, rank 3 non-breeders who were forcibly evicted from the anemone were missing. It is important to mention that we carefully inspected the areas around the focal anemone (and the anemones close by) to ensure that these individuals had actually disappeared and not dispersed elsewhere (Buston 2003, *Behav. Ecol*). They were not found elsewhere and their disappearance is almost certainly due to predation. We added this information in the revised version of the manuscript (lines 167-169; lines 178-181).

Habitat saturation

- The authors initially state that more or less every anemone is ‘full’ of fish. Thus, what you have under normal conditions is a very flat market. Even without any predation risk, it just may never pay off to search the habitat and risk losing one’s place in rank at home. I.e., you can’t ever really trade up from your ‘starter home’.

This is a good point to raise. In this system the payoff from dispersing to breed elsewhere under natural conditions has been hypothesized to be low for two reasons: i) predation risk; and ii) habitat saturation. We know that, under natural spacing of anemones, individuals don’t move even when habitat vacancies naturally occur or are experimentally created (Buston 2003 Behav Ecol; Buston 2004 Bio Let). Here, the goal was to test whether clownfish would disperse to breed elsewhere if there were habitat vacancies when the risk of movement was reduced, i.e., to determine if the fish would move when given a reasonable outside option. We find evidence that indeed they will move when risks of movement are sufficiently low.

Information availability

- Does a fish even notice that there is an empty anemone 0.5 or 5m away? The 0.5m distance would seem close enough, but fish might not even realize that a better anemone is 5m away. Therefore, they do not realize that a choice exists.

This is a good question. We chose these two distances because previous studies have shown that there are a suite of mechanisms (e.g. visual and olfactory cues) by which clownfish locate suitable reefs and hosts from considerable distances (Dixson *et al.* 2008 Proc R Soc Lond B Biol Sci, 2014 Oecologia, Fautin 1991 Symbiosis, Elliot *et al.* 1995 Mar Biol). We added this information as a rationale for the distance treatments in the revised version of the manuscript (lines 262-264).

- As a very interesting comparison, I suggest the authors look at a pair of papers by Lena Grinsted and Jeremy Field in 2017 on paper wasps, where they asked a very similar question of if subordinates will move to increase their fitness. They found subordinates do move (wasps seem more inclined to check out neighbors than clownfish), but in most cases fitness is not increased. Again, this is due to flat market, where the choices are probably very close in payoff.

These are great papers. Thank you for bringing them to our attention. We cited these into the revised manuscript (line 111).

Social constraint

My only concern here is in the verbiage used. Transplants that disappeared were claimed to be contesting for rank, and therefore ejected. The authors, however, have no data on what happened – just that fish were gone. Rather than “contest” (Lines 148-152), maybe a better phrasing is that they were “not tolerated” or “not accepted”, as there is no indication as to how the transplants behaved. Or maybe, they just left to try to get back to their original anemone?

Thank you for bringing up this major point. In this system, evictions are considered as an indication of a contest. An important point worth mentioning here is that during the third experiment, we recorded 30 minutes of video of each trial. Interestingly, the videos showed that rank 3 non breeders that were evicted from the new group did not leave the anemone trying to get back to their original anemone but, instead, they continued trying to get inside the tentacles of the new anemone, enduring the attacks of the breeders (see representative photo attached below). We considered this conduct as sign of contest for keeping a spot inside the new host/group. The fish contested for the whole length of the footage. Additionally, when

present, they were contesting even the morning of the following day when we performed the data collection.

As a minor point, clownfish were compared to social insects as species where subordinates tend to stay at home (Line 98). However, workers in many species do very often drift between nests to check them out or even join them (see Nonacs, 2017, *Frontiers Ecol. & Evol.*). They seem much less attached to their home than clownfish!

Thank you for bringing this point and the paper to our attention. Clearly we had a specific type of social insect in mind and generalized too broadly. We changed the manuscript accordingly (line 111).

Respectfully submitted,

Peter Nonacs

**Thank you for your thoughtful review, which has given us opportunity to clarify many points. Sincerely,
Rebecca Branconi**

Reviewer #2 (Remarks to the Author):

This is a well-written article on an interesting and important subject in evolutionary ecology. The authors take advantage of a great study system and use a set of experimental manipulations to make inferences.

I struggled to understand some of the key elements of the experimental designs. No doubt this is a by-product of the short format, but I think some modest elaboration is warranted. Without those details, the strict interpretation of some analyses is uncertain for the reader. The decision to not conduct a set of treatments in the first experiment was unfortunate, and raises some uncertainties around the overall interpretations.

We agree with the referee regarding the importance and necessity of explaining more in detail our experimental design and the interpretation of our analyses. We have clarified these points throughout the manuscript as suggested (see below). Regarding the decision to not conduct a set of treatments in the first experiment, please see the answer to the next comment.

Specific comments (in descending order of importance):

1. L 87-89: I think this was an unfortunate decision, and it is frustrating for the reader. I spent a fair

bit of time trying (and failing—because key details were missing) to understand the nature of these treatments, only to learn here that they were never actually implemented. While I don't dispute that movement across the longer distance is unlikely given most fish didn't move much across the shorter distance (and they didn't move 5m in Experiment 2), I think this decision represents a lapse in an otherwise elegant design. Had those trials been conducted, and no fish moved 5m in Experiment 1, the authors' inferences would have been greatly strengthened. However, had ANY of those individuals moved, this would have necessitated a substantially different interpretation. The problem is, we cannot know what would have happened, and so we must base inferences on an untested assumption (frustratingly, these treatments were originally planned, but not executed). The other issue I have with this is that it de-values the importance of robust experimental design (and serves as a poor example for others who may seek exemplars from high-impact journals). I don't think this needs to be a deal-breaker for publication, but I think some acknowledgment of the issues is warranted.

We thank the reviewer for allowing us to clarify this key point. We understand the frustration for the reader described by the reviewer but there are three main reasons why we decided to not perform another set of treatments (i.e placing alternative anemones 5 m away from the focal one) in our first experiment:

- 1. Previous studies already demonstrated that in another coral reef fish with a similar social system (*P. xanthosomus*) the likelihood of subordinates leaving the focal host to disperse to alternative ones decreased from approximately 30% to approximately 3% as distance between hosts increased from 0.1 m to 1.0 m (e.g. Wong 2010 Proc. R. Soc. B). Given that we observed only one case of movement to 0.5 m, we felt this made for a strong case for not doing the 5.0 m distance treatment. In the revised manuscript we made it clearer.**
- 2. Previous studies already demonstrated in the clown anemonefish *Amphiprion percula* that subordinates do not leave the focal host to disperse to alternative hosts when habitat vacancies are created at naturally occurring distances on the same reef (e.g., Buston 2003 Behav Ecol; Buston 2004 Bio Let). Given that we observed only one case of movement to 0.5 m and knew that there was no movement to these greater naturally occurring distances, we felt this made for a strong case for not doing the 5.0 m treatment.**

We have now communicated both points #1 and #2 in revised manuscript (lines 98-102).

2. I was unable to follow some of the experimental designs, even allowing for information split across the main text, figure captions, and supplementary methods.
Were focal anemones assigned to treatments at random?

Different classes of alternative anemones (Empty anemones or with a breeding male) were assigned randomly. We have clarified this point in lines 87-88.

Were the same anemones/fish evaluated for both distances (in the experiment where this was done)?
If so, were these done in series; was there an interval between repeated trials; was the order of presentation (of distances) randomised across focal individuals?

The same focal anemone and resident fish were used for both distances in series (0.5 meter first) and each trial lasted 2 days (no interval between trials). This was done to control for

variation among anemones and their reef context. We added this information in the revised version of the manuscript (line 119-120)

Is there any indication that individuals' responses were correlated across trials (bold/shy phenotypes)? Is there any possibility that any of these affected the outcomes of trials?

We did not test if individuals' responses were correlated across trials in relation to fish personality or any other phenotypes (e.g., size), because the focus of this study was to see if the fish would ever move. It would be very interesting to explore whether personality (or other aspects of phenotype) is predictive of movement and/or contest, i.e., explains why some fish move/contest but others do not, in future studies. We have added a paragraph on this topic at lines 216 -229.

In experiment 3, were fish cleared from anemones and then reassigned to other anemones, or was a completely new sample of 'rank 3' fish collected and used? (I ask because I wonder if precautions were taken to ensure that focal fish were not inadvertently back-transplanted to their anemone/social group of origin—which would confound the study and change my interpretations).

The introduced rank 3 individuals always came from an anemone other than the focal anemone (lines 159-161). Precautions were taken to ensure that fish were not inadvertently back-transplanted to their anemone/social group of origin by using a matrix to control the origin of the fish and the order of the introductions across anemones throughout the execution of the experiment. Additionally, each fish was collected and singularly placed inside a ziplock bag with a number indicating the tag of the anemone from which it was collected from and the number of the anemone to be transplanted to. These precautions ensured the correct execution of the experiment.

3. I think the presentation of statistical results (and their interpretation) may imply (at least to some readers) that some experimental designs were comprised an orthogonal set of treatments and independent observations. If fish were re-used, this isn't the case. Figure legends (and/or Supplementary Methods) could be modified to clarify the design and the nature of the Fisher's exact tests. In Fig 2b. the legend should more clearly indicate that the horizontal brackets indicate pairwise comparisons (as opposed to some sort of nested/hierarchical analysis—at first glance, this is how the figure appeared to me).

We thank the reviewer for this comment. We modified figure legends and supplementary methods as suggested by the reviewer to make the presentation of the statistical results and their interpretation clearer (lines 280-308)

4. L46-68: This paragraph has a structure that makes some of its components seem redundant. I suspect some minor revision could address this, to streamline the content (if necessary) and allow for more description of the breeding system (as per my next comment).

We revised the paragraph as suggested.

5. L 79-80: Some explicit predictions are offered, and to understand these, the reader needs to know about sex change in this species. This isn't explicitly covered until later. I think the breeding system could be described in the preceding paragraph (L 46-68).

We thank the reviewer for this comment. In the revised version of the manuscript we introduced the sex change of the species in the preceding paragraph as suggested (lines 63-66)

6. L 81-82: Is this certain? (e.g., because a certain size-structure was imposed by the experimental design?) The rationale for this statement isn't clear.

Yes, the reviewer is correct: a certain size-structure was imposed by the experimental design as we used breeding males individuals in the alternative anemone that were bigger than the rank 3 non-breeders of the focal anemones.

7. Line 141 and elsewhere: Consider whether the use of the term "Introducees" might cause some confusion for non-native English readers.

We thank the reviewer for this comment. We added a sentence to clarify this potential issue (line 160)

8. In several instances, "Individuals" (with a capital "I") is used; in others it is lower case. I think there may be some meaning ascribed to this, but I found it unnecessary and a distraction.

We thank the reviewer for spotting these typos. In the revised version of the manuscript "individuals" is continuously used with a lower case.

9. Line 48-49: "...avoiding inflicting..." is slightly awkward
Changed to "not inflicting"

I supply these comments in the hope that they may enable the authors to strengthen their contribution. I want to be clear that I really liked this manuscript; I think it is a worthy contribution, and I look forward to seeing it published.

We thank the reviewer for the useful comments and suggestions provided. We truly believe that the changes made in the manuscript strengthened our study.

Reviewer #3 (Remarks to the Author):

This manuscript presents the results from a set of clever and careful experiments to determine why clownfish individuals queue to wait for breeding opportunities rather than challenge for a breeding position in their own or another group. I think the data contribute important insights, the analyses are sufficient, and the findings are placed in the broader context. I have one question about the interpretation of one conclusion though, and a few suggestions for potential edits that might help readers to better understand the findings.

The interpretation I am wondering about whether it is indeed 3rd ranking males that initiate a contest or whether it is the male that would be challenged. In your experiment where you place an individual larger than 80% of the body size in a group, you argue that 12/16 of these "contested for a breeding position" (line 151). You interpret this finding saying that "we show that individuals will disperse and contest to better their current situation when ecological and social constraints are experimentally relaxed" (Line 166f). You assume that the introduced 3rd ranked individuals contest when they get closer in size - wouldn't a more parsimonious explanation be that the male starts attacking the 3rd ranking individual once it gets too close in size? I think this is an important distinction: while lower ranking individuals seem to be able to influence their growth, I am not sure they can ever switch to actively contest for that position. Whenever they grow too much, the male will start attacking and evicting them. As your results show (line 149f), these 3rd ranking

individuals never win. Accordingly it is not a true contest, it rather seems that males feel challenged and evict. This also makes me wonder how comparable this situation is to the economic bargaining theory that you describe: in it, actors appear to have a choice because individuals seem to be relatively similar. In the clownfish, there are inherent differences among individuals that limit the options individuals might be able to take. Do you have other data that might put these conclusions in context? For example, is contest based on a probabilistic assessment, where the closer in body size 3rd ranking individuals are the more likely they will be attacked?

We thank the reviewer for bringing up this point. As explained in response to comment of the first reviewer, here, as in previous studies, evictions are considered as an indication of a contest between individuals — the subordinates persist in engaging in the contest rather than walking away. Importantly, previous studies of clownfish showed that dominants never lose a contest with their subordinates when the size ratio between them is less than the 0.8 found under natural conditions. However, they suffer a significantly higher risk of losing the contest when the size of the subordinate is closer or above that ratio (Buston & Cant 2006 *Oecologia*; Wong et al 2016 *Behav Ecol Sociobiol*). So, 3rd ranked individuals are capable of influencing their growth, contesting for higher positions, and winning contests against their immediate dominants when their size is similar.

Some additional information that might be helpful:

You mention that there is a "high recruitment rate" (Line 55): it might help to have more information on how recruitment occurs (if you have it): is it larval and they crawl, do individuals float and randomly settle on the first anemone they find, or do they swim within a certain radius?

In clownfish, eggs are laid next to the base of the anemone, cared for by the parents for 7 days until they hatch (Barbasch & Buston 2018). On hatching, the larvae disperse from their natal anemones a mean distance of about 15 km and up to 120 km (Almany et al. 2017 *Nat Eco Evo*). Following the dispersing larval phase, they find a reef using a variety of cues and locate suitable anemones using chemical and visual cues (Elliott et al. 1995 *Mar Bio*). There is a constant rain of these larval settlers down on to the reef, but settlers are only allowed to enter anemones with a low degree of saturation (Buston 2003 *Behav Ecol*).

For the individuals displaced 5 meters to a new anemone, you say that none returned to their home anemone (line 121): what happened to them? Did they move at all? Did they die?

They didn't move at all. They were all found inside the alternative anemone. We added this information in the revised manuscript (lines 140-141).

Are dominant males/females all of the same size, independent of which anemone they are in? If there is variation, would a 3rd ranking individual from a large anemone be able to outcompete a male/female from a smaller anemone?

There is a huge variation in the size of the dominant males and females across different groups. This is because the size of the dominant individuals depends on the size of their anemone: bigger is the host, larger is the fish group size and bigger are all the individuals within the group (Godwin & Fautin 1992 *Copeia*; Buston 2003 *Nature*). Potentially, a 3rd ranking individual from a larger anemone would be able to outcompete a male/female from a smaller anemone (Wong et al. 2016 *Behav Ecol Sociobiol*). However, as shown in this study, this does not occur in this species because individuals normally do not move between different anemones.

But this might not occur because, as you show, there is some form of home advantage (knowing the place, the other individuals) or because their own fitness will be so much larger on the better anemone that it makes up for waiting?

Regarding the “home advantage” (or benefits of philopatry hypothesis) mentioned above by the reviewer, future studies are certainly needed to assess exactly what are the benefits of remaining in the same host and within the same group. In fact, there is a huge variability in the quality of the host and of the other fish that can influence non-breeders’ decisions. In the revised manuscript we added a paragraph to better clarify all these points (lines 201-215).